# Role of Dentistry in Humanitarian Projects: Knowledge and Perspective of Future Professionals on the World of Volunteering in Spain

**DOI:** 10.3390/healthcare11071055

**Published:** 2023-04-06

**Authors:** Matías Ferrán Escobedo Martínez, Enrique Barbeito Castro, Sonsoles Olay, Brezo Suárez-Solis Rodríguez, Juan Suárez-Solis Rodríguez, Luis Junquera, Mario Mauvezín Quevedo, Sonsoles Junquera

**Affiliations:** 1Department of Integrated Adult Dentistry, School of Dentistry, University of Oviedo, C/. Catedrático Serrano s/n., 33006 Oviedo, Spain; 2Department of Gastroenterology, Hospital Universitario Central de Asturias, 33004 Oviedo, Spain; 3Head Department of Oral and Maxillofacial Surgery and Oral Medicine, School of Dentistry, University of Oviedo, C/. Catedrático Serrano s/n., 33006 Oviedo, Spain; 4Department of Radiology, Hospital Universitario de San Agustin, 33402 Aviles, Spain

**Keywords:** global oral health, dental aid organization, non-governmental organization, dental NGO, oral health development, volunteer, basic oral care, primary health care

## Abstract

Non-governmental organizations (NGOs) in dentistry seek to promote the improvement on oral health in the most disadvantaged regions. The objective of this study is to identify the level of knowledge, expectations, and motivations that dental school students have about volunteering in dentistry, as well as to evaluate possible differences in these variables depending on their level of dental training. During the month of September 2022, a voluntary and anonymous online survey was carried out among all the students at the Dentistry School of Oviedo University. There were 5 questions to judge knowledge about global oral health course. 12 additional questions were included to assess the willingness to volunteer in international setting, the volunteer profile, as well as the most effective means to improve oral health in host communities. None of the students from our center had participated as a volunteer in dental NGOs, but up to 64.4% of them had considered their collaboration. The level of knowledge about global oral health obtained was low, with the percentage of correct answers ranging between 14.4% (in the question about the ideal patient/dentist ratio) and 57.8% (in the question about the fluoride concentration in drinking water). Majority of dental students (98.9%) were not aware that basic package of oral care was created by WHO. Significantly, the students of the clinical courses showed a greater motivation to volunteer.

## 1. Introduction

The term non-governmental organization (NGO) is very broad and encompasses many different types of organizations. They include international charities, research institutes, churches, community-based organizations, lobby groups and professional associations. Traditionally, NGOs are value-based organizations that depend in whole or in part, on charitable donations and voluntary service [1,2]. In 1950, The United Nations (UN) Department of Public Information defines the non-governmental organization (NGO) as *“a not-for profit, voluntary citizen’s group that is organized on a local, national, or international level to address issues in support of the public good. Task-oriented and made up of people with a common interest, NGOs perform a variety of services and humanitarian functions, bring citizen’s concerns to Governments, monitor policy, and program implementation, and encourage participation of civil society stakeholders at the community level”* [3].

NGOs are part of what is called the third sector or social sector or non-profit sector. Currently, it is impossible to assess the accurate number of NGOs neither on a global scale, nor countries. It is related to many organizations, a variety of legal forms of entities and their registration, as well as the dynamics of their formation and stoppage of activity of the organizations [4,5]. 

This study focuses on those organizations that are dedicated to health activities that aim to improve the physical and mental state of a population [6]. According to the UN, this type of organization represents almost a third of the total number of NGOs currently in existence, with three standing out from all of them due to their greater public recognition: “*Dentists without borders*”, “*Medicus Mundi*”, “*Partners in Health”* [7,8,9,10]. 

Oral health is an essential component for a good state of general health and is considered a fundamental human right. The World Dental Federation (FDI) defines oral health as *“Oral health is multi-faceted and includes the ability to speak, smile, smell, taste, touch, chew, swallow and convey a range of emotions through facial expressions with confidence and without pain, discomfort and disease of the craniofacial complex (head, face, and oral cavity)”* [11].

The FDI developed a roadmap of the challenges that would have to be met to achieve “optimal and equal” oral health worldwide by 2030 [12]. The challenges to be met include: (1) By 2030, essential oral health services are integrated into healthcare in every country and appropriate quality oral healthcare becomes available, accessible, and affordable for all. (2) By 2030, oral and general person-centered healthcare are integrated, leading to more effective prevention and management of oral diseases and improved health and well-being. (3) By 2030, oral health professionals will collaborate with a wide range of health workers to deliver sustainable, health-needs-based, and people-centered healthcare.

The traditional approach applied to oral health has been directed mainly towards treatment rather than prevention and promotion of oral health [13]. Worldwide, oral diseases are the fourth most expensive pathology to treat and, according to data published in the Global Burden of Disease Study in 2019, they would affect nearly 3.5 billion people worldwide [14,15]. Currently, caries is the most frequent disease, and it is estimated that 2000 million adults suffer from it (permanent teeth) and 520 million children (decidual teeth) [15]. Another important point about the state of oral health is periodontal disease. According to the World Health Organization (WHO), it is estimated that periodontal disease affects almost 10% of adults, which represents more than 800 million cases worldwide, being poor oral hygiene and tobacco the main causes of this disease [16]. Along with these data, we must highlight that today there are great inequalities in access to oral care between developed and non-developed countries: for example, in 2020 in Croatia and Spain, the ratio of dentists per inhabitant amounted to 1:560 and 1:1171 respectively, while in Ethiopia it barely reaches 1:1,278,000 [17].

Other factors to take into account in oral health are the level of education and social status. In 2017 Chaffee et al. [18] published a study comparing the development of caries in childhood in relation to the socioeconomic and educational status of their families. A total of 456 children were analyzed, 60% of the untreated caries had developed in children from families with a low or very low educational level and few economic resources. Guarnizo-Herreño et al. [19] recently published a study in Colombia (7877 adults) that analyzed edentulism and caries development in relation to socioeconomic and educational level. This study showed that the population with university studies and with 3 or 4 times the monthly national minimum wage (NMW) (high economic level) presented 3 times less untreated caries and 4 times less edentulism than the population that had only completed education primary and half of NMW (low income). These two works demonstrate the importance of these two factors in the state of oral health and reinforce the need to focus state aid and volunteering on the most vulnerable socioeconomic stratum.

Recently, there has been a growing international awareness of the idea that oral health constitutes a fundamental part of general health, with an extremely positive movement towards the inclusion of oral health within general health plans [20,21]. In 2021, the WHO approved a resolution recommending the inclusion of oral health in chronic disease prevention programs [20,22]. The current problem arises when trying to equalize the oral health status of underdeveloped countries with that of developed countries, given the significant difference in resources allocated for this purpose in each of them. This imbalance is where dentistry NGOs come into play, promoting the improvement of oral health in the most disadvantaged regions.

The aim of this exploratory study was to identify the level of knowledge, expectations, and motivations about volunteering in dentistry that the students at the School of Dentistry of the University of Oviedo (Spain) might have, as well as assess possible differences between them based on their level of dental training. Secondarily, it is intended to evaluate the knowledge of students about the main indicators of oral health in the world population.

## 2. Materials and Methods

### Study Design

An observational cross-sectional study was carried out, through a survey (see Appendix A) to the students at a Dentistry School in the north of Spain, using the virtual platform “GOOGLE FORMS”. All the surveys were anonymous, voluntary, and were carried out to all the students of the faculty (between the first and fifth grade courses) during the month of September 2022. The survey used in this study is original, however the questions about world oral health took as reference the studies of Singh and Purohit et al. [23] and Braimoh and Odai et al. [24]

The survey was divided into 3 parts:1.Demographic data:

This first section was made up of 3 questions that sought to find out the sex, age group and academic year of the dental students surveyed. (See Appendix A)

2.Questions about the knowledge of global oral health:

The second section included 5 questions. 4 questions sought to find out what the students knew about the worldwide prevalence of untreated caries and periodontal disease, the recommended range of patients per dentist as well as the fluoride concentration recommended in public waters by the WHO [25]. All these questions had only one correct answer (see Appendix A “questions about the knowledge of global oral health” correct answer in red colour).

The last question asked if the respondents had ever heard of the Basic Oral Care Package (BPOC) developed by the World Health Organization (WHO) for the prevention and treatment of dental pathologies in public health systems [26].

3.Questions about the general and specific knowledge of dental volunteers:

The last and third section consisted of 12 opinion questions about volunteering.

The first two questions were intended to find out if the respondents had ever proposed or had already volunteered, whether at a dental or non-dental level.

The third question (multiple response option) was presented by a total of 10 dental NGOs to find out their degree of recognition by those surveyed.

In the fourth, fifth and sixth questions, the aim was to find out the opinion of those surveyed about their willingness to pay for volunteering, the range of financial outlay they were willing to pay, as well as whether they would volunteer in a country in conflict (wars, dictatorships…).

The seventh question (multiple response option) asked who they believed to be the most representative volunteer dentist profile. Three types of profiles were presented: new dentist “5 years working”; senior dentist “more than 5 years working”; retired dentist “retired person”.

The eighth and ninth questions (multiple-response option) sought to find out the opinion of the respondents on the motivations for volunteering, as well as the most necessary dental treatments that should be carried out in a dental volunteer.

The tenth question (multiple response option) was a survey about which regions of the world they believed were most in need for volunteering.

Finally, the last two questions in this block wanted to know, on the one hand, the age group (children or adults or old people) that most needed help from dental NGOs and, on the other hand, the most important actions to improve oral health of the areas where dental volunteering is carried out.

Since this survey was designed to find out the opinions of those surveyed, it was only possible to evaluate and score knowledge of 4 of the 12 questions included in the survey (questions Section 2: world prevalence of untreated caries and periodontal disease, the recommended range of patients per dentist as well as the fluoride concentration recommended in public waters by the WHO). It was considered that the knowledge about these questions was low if the respondents did not exceed 50% of correct answers in each question, medium up to 70% of the correct answers and high more than 90% of the correct answers.

The respondents were divided into two groups. The first of them was made up of the preclinical courses (1st and 2nd year of dentistry) and the second one by the clinical group (3rd, 4th, and 5th year of dentistry).

The data obtained from the different variables, duly coded, were entered into a database, using the R statistical software (R Development Core Team, version 3.6.3, Vienna, Austria) for data analysis. A descriptive analysis was performed, providing absolute and relative frequency distributions for the qualitative variables, and for the quantitative variables position and dispersion measures were provided. The relationships between variables were studied with Pearson’s Chi-square test or with Fisher’s test depending on whether or not the hypothesis about expected frequencies is verified. Statistically significant differences were considered those variables in which the *p* value was less than 0.05 (*p* < 0.05).

## 3. Results

Of the 96 students enrolled in the 2022–2023 academic year at our Center, 90 students completed the surveys of this study (93.7% of the total). If we look at the academic courses, we can see that 16.6% correspond to the first year, 17.8% to the second year, 18.9% to the third year, 20% to the fourth year and finally 26.7% corresponded to the students of the last year. The majority (77.8%) of those surveyed were between 18–25 years of age, with 73.3% of the total being female. The second part of our survey reflected the general knowledge that respondents had about “global oral health” and was structured around 6 main questions:

The first question asked about the global prevalence of untreated caries (20–30% prevalence; WHO 2022) [25], only 26.7% answered this item correctly (Table 1).

The second question was about the global prevalence of periodontal disease, answering the correct option (10–15% prevalence; WHO 2022) [25] only 20% of respondents (Table 2). 

The third question was intended to assess the knowledge that respondents had about the “basic package of oral care (BPOC)” created by WHO [26]. Majority of dental students (98.9%) were not aware that basic package of oral care (BPOC) was created by WHO. None of the surveyed students correctly answer the question “name the three components of BPOC”.

According to the WHO [25], the ideal ratio of patients per dentist should be 3500 patients/dentist; only 14.4% of the students answered this question correctly.

The last question in this block asked about the amount of fluoride recommended in drinking water by the WHO for caries prevention (0.5–1 mg/L) [25]. About 60% (57.8%) of the respondents answered the question in the right way.

In the third section of the survey, the opinion of the students at our University about “dental volunteering” is collected:

In this regard, they were asked if they had ever considered doing some type of volunteering (not necessarily related to their profession), 76.7% answered affirmatively. The following question asked the same question as in the previous question, but specifically in the dental field, with 64.4% responding affirmatively (Table 3 and Table 4). 

The 10 most Influential NGOs in the Dental Field, respondents placed “*Dentists without borders*” in first place (65.5%), followed by “Solidarity Dentistry” (20%) and “SMILES” in third place (16.7%) (Table 5).

54.4% of the students would be willing to pay to carry out voluntary actions, with an economic expense threshold of less than €800 in most cases (65.6%), between €800–1500 in 27.8% and only more than €1500 in 6.7% of them.

If the sociopolitical situation of the country were conflictive (wars, dictatorships, serious political instability) only 37.8% of the students would be willing to provide volunteer services there.

In the next item, they were asked about the professional profile that they believed was most frequently interested in dental volunteering, being the new dentist profile the one they believed to be most frequently involved (91.1% of the surveys), followed by the senior (21.1%) and in last place the profile of retired dentist (17.8%).

Considering the different intentions that motivate volunteers to carry out their projects, those surveyed considered that the most frequent motivation is to help people without resources (95.6%), followed by learning about new cultures and lifestyles (67.8%), improving their own professional skills (62.2%) and lastly ideological/religious reasons (4.4%).

According to those surveyed, the dental treatments most required within the dental volunteers are conservative in the first place (97.8%), followed by periodontics (64.4%), prosthodontics (43.3%), oral surgery (36.7%) and lastly orthodontics (10%).

The most effective preventive strategies to improve oral health according to dental students would be education for the population (94.4%), provision of material resources (86.7%), provision of care services by volunteer dentists (65.6%), training new professionals in the country (43.3%) and finally the promotion of the creation of new care centers (27.8%).

They were questioned about the geographical regions where they considered there was a greater need for humanitarian actions in the dental field, with sub-Saharan Africa (92.2%) being the most frequently answered option, followed by Latin America (70%), Western Asia (51.1%) and North Africa (40%). Those surveyed considered that the age group most in need of care would be children (49% of responses), closely followed by adults (40%) and the elderly in last place (11%).

Finally, we analyzed the differences between the responses of students with clinical training (students from 3rd to 5th year) compared to students with basic/preclinical scientific training (students from 1st and 2nd year), observing statistically significant differences in the following sections:

Worldwide prevalence of untreated caries: A higher number of correct answers was observed in favor of the preclinical group (*p* = 0.012) (Table 6).

Age group that needs assistance: Students in the preclinical group considered that the elderly would require more health care than children compared to the clinical group (*p* = 0.029) (Table 7).

Knowledge of the different dental NGOs: The preclinical group knows more frequently the organization “SMILES” and the clinical group the organization 

“*Dentists without borders*” (*p* = 0.002 and *p* = 0.024 respectively) (Table 8 and Table 9).

Motivation to volunteer in dentistry: Students in the clinical group considered that “improving professional skills” is one of the most frequent motivations for volunteering compared to the preclinical group (*p* = 0.028). (Table 10).

Strategies to improve oral health in the populations treated: the preclinical group considered that the training of new professionals in the country represented a better prevention strategy compared to the students in the clinical group. (*p* = 0.023) (Table 11).

No statistically significant differences were observed between the different groups in the other items.

## 4. Discussion

Although previous studies have found that interest in dental volunteering is growing [27], the results of our survey showed that none of the respondents had participated in volunteer activities related to oral health.

Two of the most prestigious scientific journals in the world have very recently emphasized the importance of oral health. In the first of these studies carried out by D’Souza et al. [15], the issue was analyzed in the United States of America (USA), highlighting the importance of oral health for general health, capturing the attention of researchers, legislators, professionals, and the general public. Moreover, the health systems of developed countries recently began to recognize and quantify the inequalities in oral health that affect marginalized populations, due to complex structural and interpersonal variables such as racism, xenophobia, or religious fundamentalism. Thanks to this, slow implementation solutions are beginning to be put in place.

The second article on the importance of oral health was published in 2021 by Benzian et al. [28]. In this study, they focus on the well-known neglect of oral health on the global health agenda, that is the reason why the resolution on oral health promoted by the WHO in 2021 is considered an important advance [20]. 

The Lancet Commission on Oral Health [28] includes a series of Key recommendations for the new WHO global strategy for oral health: 1. Inclusion and community engagement 2. Place equity and social justice at the core 3. Tackle sugars as a major common risk factor 4. Embrace major system reforms 5. Better data for decision making 6. Close financing gaps. In addition to these six recommendations, some experts include a seventh that would be based on the communication of evidence-based experiences and the exchange of skills between doctors and dentists.

Today more than 70% of the world’s population, mainly those living in low-and middle-income countries, have little or no access to oral health care. Although oral health is recognized as a basic human right, the lack of appropriate and affordable oral care to more than 4 billion people worldwide does not result in a massive increase of political activity and financial resources to address the problem [28,29].

Benzian et al. in a study to collect basic data about non-governmental dental aid organizations on a global scale reported that about two thirds of NGOs originated in industrialized countries and one third responded from developing countries. The majority had been established after 1980 [30].

However, and despite a growing importance of NGOs in the medical and general health sector, which has brought about a new generation of highly professional, socially responsible, and financially transparent organizations, the situation in the sector of oral health development assistance is very different [30]. The sector is characterized by at least six deficiencies: 1. It is rather small (with an assumed maximum of 1000 NGOs operating worldwide). 2. The financial resources for many NGOs are very limited. 3. The degree of professionalism is generally very low. 4. Curative approaches based on technical interventions and service provision dominate. 5. Integration into existing local community structures is often very low. 6. A lack of coordination, information and technology sharing between the different dental NGOs. Despite these limitations interest in international volunteering has extended from practicing dentists to dental schools. In response to the increased interest in global oral health care among students, dental schools in the USA have expanded opportunities for students to engage in clinical care in international settings [29] [31]. However, in Spain, very few dental schools have taken this type of initiative. 

Lambert et al. [29] conducted an exploratory study through a survey, to find out the interest and experience of dental pre-doctoral students in global health. The survey sought to assess students’ past experiences and current and future interest in programs that provide dental and/or medical services in order to lay the groundwork for future research. The survey consisted of 12 questions evaluating students’ experiences and perceptions about international volunteering/service trips. Survey questions included student nationality, whether the student had experienced a service trip (yes/no), length of service trip(s) (list of choices from <1 week to >6 months), preparation for service trip(s) (Likert scale questions), and impact of service trip(s) (Likert scale questions). Students were also asked to provide motivations for and barriers to international service trip involvement (choice of a list of options). Participants were asked to select the option corresponding to their most recent volunteer experience and their longest volunteer experience if they had been on more than one service trip.

A total of 1555 dental students responded out of 22,930 eligible students, resulting in a response rate of 7%. Of respondents who reported participation in service trips (342, 22%) abroad, 66% had only one such experience, 26% had participated in two or three experiences, and 8% reported participation in more than three trips. 56% of the respondents reported participating in service trips sponsored by their dental schools as opposed to external organizations. This is an important difference with our university, where there are no humanitarian health programs in dentistry, which perhaps explains why none of the respondents to our work have participated in volunteer work. Nevertheless, the motivations for participating in this type of activity were similar among the respondents of both studies. Of the 1213 study participants who had not participated in an international service experience in Lambert el al. study [29], 72% reported that their school offered international dental service trips, 16% stated that their school did not offer such experiences, and 12% were unsure. Although this group had not yet experienced a service trip, 83% reported interest in participating while in dental school. For those interested in service trips, helping the underserved, gaining a global view of health, care, and disease, being exposed to new experiences, and improving clinical skills served as major motivators.

In our study, 54.4% of the students would be willing to pay to carry out volunteer actions. If the country’s sociopolitical situation were conflictive (wars, dictatorships, serious political instability) only 37.8% of students would be willing to volunteer there.

However, in other surveys [29] concerns for those who had not yet experienced an international service trip but were undecided whether they would participate in one after dental school were cost/lack of funding (67%), logistical challenges (37%), lack of opportunities (21%), and safety concerns (21%); however, time constraints were more of a barrier, with 67% selecting that option. 

Our School of Dentistry does not have any tradition of volunteering as an institution and to our knowledge this is the first study carried out in Spain on this issue among dental students. 64.4% of the students had considered participating in dental volunteering on some occasion. Although it is a high percentage, it is much lower than that observed by Lambert et al. [29] in his survey in the USA. In this study, 83% reported interest in a service trip while in school, and 92% were interested after graduation. Reported motivations for international trips included the desire to care for the underserved and to obtain a more global view of health and disease. 

It is important to know the motivations of students to participate in volunteer programs. Many volunteer programs focus entirely on the success and happiness of the volunteers, often at the expense of the host community [32]. The most problematic motivations include selfishness, feelings of adventure and a way to gain clinical exposure [33]. 

Although, most of the studies agree with our results regarding the positive motivations of volunteer participants, few studies make comparisons between the level of professional training of those surveyed Lambert el al. study [29]. Our study observed the existence of significant differences in the motivations for volunteering. Students in the higher grades considered that “improving professional skills” is one of the most frequent motivations for volunteering compared to the preclinical group. For other authors, the motivations, and barriers for health professional students to volunteer on an NGO may depend on the context of the trip and the opportunities it provides, rather than an intrinsic desire to volunteer. Demographic differences between volunteers may be helpful in designing motivation to volunteer on a medical service trip [34]. 

The level of knowledge about global oral health obtained in our study was low. Although for other authors, the students surveyed reported feeling prepared or very prepared for volunteering activities, reports from the communities that receive care often state that volunteers are not prepared to provide health care services in their specific situation [29,35]. 

Our respondents, mostly women, were largely unaware of the global prevalence of untreated caries and periodontal disease established by the WHO [25]. On the other hand, although 60% of the respondents knew the percentage of fluoride in drinking water recommended by the WHO [25], only 14.4% of the students correctly answered the value of the ratio recommended by the WHO of patients/per dentist and a minimum percentage (1.1%) recognized the BPOC concept and the three parts of which it is composed.

Braimoh et al. in Nigeria [24] investigated the preparation of students to understand the status of oral health conditions globally. Half of the surveyed students stated that their dental education has “moderately” prepared, especially in developing countries. Nearly 27.3% of the surveyed students stated that they were “somewhat” prepared, and 22.7% percent stated that they were “greatly” prepared by their dental education to understand the status of oral health conditions globally. Although knowledge of the basic package of oral care (BPOC) was low in this study (4.5%), it was much higher than that observed in our survey (1.1%).

On the other hand, a survey of 3487 dental students at eight schools in seven countries showed that volunteerism and philanthropy are important qualities of a well-rounded dentist, but only about a third of those surveyed felt that their school supported these behaviors (36.2%) or demonstrated a commitment to promoting global dentistry (35.5%). Additionally, 87.4% felt that dental schools are morally obligated to improve oral health care in underserved global communities and should provide students with volunteer assignments [36].

This study is limited by several factors. In first place, because of the small sample size, and in the other hand, as with all survey studies, the responses may not be representative of all dental schools in Spain. However, results provide exploratory and fundamental information knowledge from which interventions and more research can be developed. We believe that for future studies, a multicenter study should be carried out to compare the knowledge and attitudes of dental students from different universities in Spain and Latin America about NGOs.

## 5. Conclusions

In conclusion, in recent years the importance of oral health has been revalued internationally, as a fundamental part of general health. However, at the present time, medicine and dentistry remain separate worlds as far as NGOs are concerned. Although dental students showed a predisposition to participate in dental volunteering for different reasons, their knowledge on global oral health issues and on the strategies that dental NGOs should develop was insufficient. There is a training gap in our School on this issue, so changes should be made to the program on prevention, sustainability, and maintenance of oral care in the most disadvantaged communities.

In a well-defined global context and appropriate learning environments that are respectful of host communities’ goals and efforts, dental students’ interests and enthusiasm provide a tremendous opportunity for the dental profession to positively influence health at the global level for generations to come.

## Figures and Tables

**Table 1 healthcare-11-01055-t001:** Percentage of responses about the global prevalence of untreated caries (correct answer red colour).

	FREQ.	%
10–25% *	10	11.1
20–30% *	24	26.7
50–60% *	56	62.2
TOTAL	90	100.0

* untreated caries prevalence range, FREQ. = number of students for each answer.

**Table 2 healthcare-11-01055-t002:** Percentage of responses about the global prevalence of periodontal disease (correct answer red colour).

	FREQ.	%
3–8% *	2	2.2
10–15% *	18	20.0
25–30% *	45	50.0
45–60% *	25	27.8
TOTAL	90	100.0

* Periodontal disease prevalence range, FREQ. = number of students for each answer.

**Table 3 healthcare-11-01055-t003:** Answers about general volunteering.

GENERAL VOLUNTEERING		
·	FREQ	%
NO	19	21.1
YES	69	76.7
I HAVE ALREADY DONE A VOLUNTEER	2	2.2
TOTAL	90	100.0

FREQ. = number of students for each answer.

**Table 4 healthcare-11-01055-t004:** answers about dental volunteering.

DENTAL VOLUNTEERING		
	FREQ	%
NO	32	35.6
YES	58	64.4
I HAVE ALREADY DONE A VOLUNTEER	0	0
TOTAL	90	100.0

FREQ. = number of students for each answer.

**Table 5 healthcare-11-01055-t005:** Percentage of respondents who know the NGOs.

NGOs	RESPONDENTS WHO KNOW THE NGOs (%)
ODSOLIDARY	8.9
DENTISTS WITHOUT LIMITS	6.7
DENTISTS WITHOUT BORDERS	65.6
DENTAL COOP	7.8
DENTISTS ON WHEELS	4.4
SOLIDARITY DENTISTRY	20.0
PLANETARY ACTION	5.6
SMILES	16.7
CLOSE AND FAR	5.6
SOS SOCIAL DENTISTRY	4.4
I don’t know any dental volunteer NGOs	0
TOTAL	100.0

**Table 6 healthcare-11-01055-t006:** Differences between groups in the prevalence of untreated caries (*p* = 0.012) (correct answer red colour).

PREVALENCE OF UNTREATED CARIES (20–30% Prevalence; WHO 2022) [25]						
GROUP	10–25% *		20–30 % *		50–60% *	
	FREQ.	%	FREQ.	%	FREQ.	%
PRECLINICAL	6	19.35	12	38.71	13	41.94
CLINICAL	4	6.78	12	20.34	43	72.88

* untreated caries prevalence range. FREQ. = number of students for each answer.

**Table 7 healthcare-11-01055-t007:** Differences between groups regarding the age group that needs more assistance (*p* = 0.029).

AGE GROUP THAT NEEDS ASSISTANCE						
GROUP	CHILDREN		ADULTS		OLD PEOPLE	
	FREQ.	%	FREQ.	%	FREQ.	%
PRECLINICAL	11	35.48	13	41.94	7	22.58
CLINICAL	33	55.93	23	38.98	3	5.08

FREQ. = number of students for each answer.

**Table 8 healthcare-11-01055-t008:** Differences between groups about knowledge of the NGO “SMILES” (*p* = 0.002).

KNOWLEDGE OF THE DIFFERENT DENTAL NGOs “Smiles”				
GROUP	NO		YES	
	FREQ.	%	FREQ.	%
PRECLINICAL	20	64.52	11	35.48
CLINICAL	55	93.22	4	6.78

FREQ. = number of students for each answer.

**Table 9 healthcare-11-01055-t009:** Differences between groups about knowledge of the NGO “Dentists without borders” (*p* = 0.024).

KNOWLEDGE OF THE DIFFERENT DENTAL NGOS “Dentists without Borders”				
GROUP	NO		YES	
	FREQ.	%	FREQ.	%
PRECLINICAL	16	51.61	15	48.39
CLINICAL	15	25.42	44	74.58

FREQ. = number of students for each answer.

**Table 10 healthcare-11-01055-t010:** Differences between groups regarding the motivations for volunteering (*p* = 0.028).

MOTIVATION TO VOLUNTEER IN DENTISTRY				
GROUP	NO		YES	
	FREQ.	%	FREQ.	%
PRECLINICAL	17	54.84	14	45.16
CLINICAL	17	28.81	42	71.19

FREQ. = number of students for each answer.

**Table 11 healthcare-11-01055-t011:** Differences between groups on the strategies to improve oral health in the populations intervened. (*p* = 0.023).

STRATEGIES TO IMPROVE ORAL HEALTH IN THE POPULATIONS TREATED				
GROUP	NO		YES	
	FREQ.	%	FREQ.	%
PRECLINICAL	12	38.71	19	61.29
CLINICAL	39	66.10	20	33.90

FREQ. = number of students for each answer.

## Data Availability

Not applicable.

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
