# Peer review of "Role of Dentistry in Humanitarian Projects: Knowledge and Perspective of Future Professionals on the World of Volunteering in Spain"

_healthcare, 2023, doi:10.3390/healthcare11071055_

Round 1
Reviewer 1 Report
Please find the attachment

Author Response
Dear Authors,
Thanks for this good job, I found the topic interesting. Following you can find some feedback:
- In introduction section, it would be better to add few studies regarding the effect of level of education and social status on this matter. Thank you very much for the advice, we have corrected it ( see lines 84-96)
- In method and material section line 113-114: you can rewrite it to (1st and 2nd year of dentistry/dental school) for better understanding. we have corrected it
- Including more dental schools and therefore students could lead the study to a multi-centerstudy with more reliable results. Maybe you can recommend it for the future studies. Thank you very much for the advice, we will take it into account for future studies.
- Line 120: please rewrite it for a better understanding. 90 Student! we have corrected it
Tables: please make some changes and add more information to the tables. For example It’s not defined in the table1, what is the 10-25,…?
we have corrected it see:
lines 193-201(table1)
lines 206-213 (table 2)
lines 233-251 (table 3-4)
lines 317-318 (table 6)
lines 324-325 (table 7)
lines 331-332 (table 8)
lines 336 (table 9)
lines 342-343 (table 10)
lines 350-353 ( table 11)
- In Discussion section, you can add more interpretation comparing the result of your studieswith others.
we have corrected see the discussion section
- Please add the limitation of your study and your suggestions for the future studies.
we have corrected it see Lines 501-507

Reviewer 2 Report
I would like to congratulate the authors for this study, but I would like to suggest some improvements.
I suggest the authors to summarize the text and to redo the discussion section, emphasizing the results and try to explain, discuss and justify the findings
Author Response
I would like to congratulate the authors for this study, but I would like to suggest some improvements.
I suggest the authors to summarize the text and to redo the discussion section, emphasizing the results and try to explain, discuss and justify the findings
we have corrected see the discussion section

Reviewer 3 Report
Some editing of written English is required for clarity. Paragraphs should focus on one topic with a topic sentence. Create new paragraph when changing topic.
Methods section required more description of the survey questions. Explain to reader how the questions were asked and answered, multiple choice, etc. Was the survey given in Spanish or English? The question about what NGOs respondents are aware of is particularly confusing. Are you testing knowledge or just awareness? Define all categories of responses - for example, what is meant by new dentist profile, senior, and retired dentist. How are these defined? Do not expect reader to get this information from the appendix. Also explain how results are scored. Most importantly, what statistical tests were applied. You mention a p-value for significance, but don't report how it is obtained or what the specific results are. Statistical significance is mentioned but not shown in tables.
Consolidate tables. What are the demographic characteristics of those surveyed? One table should show all descriptive statistics, and another for results of other statistical tests, along with p values. All questions asked should be described in the methods, or in the results tables, as they were worded and including what the responses could be (if multiple choice).
Much of the discussion belongs in the introduction. The beginning paragraphs relating to the United States aren't relevant. If included, it should be shortened. When discussing disparities, those between nations are more relevant to your research than those within nations. The paragraph on page 10 from lines 353 to 369 seems to contradict your argument rather than support it. The question and answer section beginning on page 10 is not an appropriate format for a scholarly article and also doesn't relate to your research. This section reads like a student essay and doesn't belong in this article.
Your appropriate discussion appears to begin on page 11 line 433. Do not simply repeat results in discussion, explain them and their impact. What are the implications for the future of dental education? The future of humanitarian efforts? Oral health disparities overall? For your conclusion, make sure it is supported by your results, what changes should be made in dental education based on your results?
If including the entire survey as an appendix - consolidate to as few pages as possible by eliminating all extra spacing.
Author Response
Was the survey given in Spanish or English? The survey was given in Spanish
Statistical significance is mentioned but not shown in tables.
we have corrected it see:
lines 317-318 (table 6)
lines 324-325 (table 7)
lines 331-332 (table 8)
lines 336 (table 9)
lines 342-343 (table 10)
lines 350-353 ( table 11)
Methods section required more description of the survey questions.
we have corrected see the material and methods section
Explain to reader how the questions were asked and answered, multiple choice, etc.
we have corrected see the material and methods section
The question about what NGOs respondents are aware of is particularly confusing.
we have corrected see the material and methods section
Define all categories of responses - for example, what is meant by new dentist profile, senior, and retired dentist. How are these defined? Do not expect reader to get this information from the appendix.
we have corrected see the material and methods section
Also explain how results are scored.
we have corrected see the material and methods section
Most importantly, what statistical tests were applied. You mention a p-value for significance, but don't report how it is obtained or what the specific results are.
we have corrected see the material and methods section
Much of the discussion belongs in the introduction. The beginning paragraphs relating to the United States aren't relevant. If included, it should be shortened. When discussing disparities, those between nations are more relevant to your research than those within nations. The paragraph on page 10 from lines 353 to 369 seems to contradict your argument rather than support it. The question and answer section beginning on page 10 is not an appropriate format for a scholarly article and also doesn't relate to your research. This section reads like a student essay and doesn't belong in this article. Your appropriate discussion appears to begin on page 11 line 433. Do not simply repeat results in discussion, explain them and their impact. What are the implications for the future of dental education? The future of humanitarian efforts? Oral health disparities overall? For your conclusion, make sure it is supported by your results, what changes should be made in dental education based on your results?
we have corrected see the discussion section
If including the entire survey as an appendix - consolidate to as few pages as possible by eliminating all extra spacing.
we have corrected it see Lines 537-638

Reviewer 4 Report
This article entitled role of dentistry in humanitarian projects. knowledge and perspective of future professionals on the world of volunteering in Spain aims to analyze the level of knowledge about the level of general health and the motivations for volunteering in dentistry.
Materials and method
- Review the score in the study design section.
- It could be interesting in this section to give a summary of each of the three blocks of questions to give a general idea of ​​them.
Results
- We recommend reviewing the names of the NGOs since some are translated into English and others are not. the same criteria should be followed
Discussion and conclusions
- As the authors have commented, the questionnaire was carried out at the beginning of the course (September 2022), so the first-year students have just entered the degree. Were the differences between newly admitted students and those who had been a year old (second-year students) assessed?
- After the results obtained and the comparison with previous studies, what would be the future for this field of analysis?
- Authors are recommended to review this section and include more analysis of the results obtained as well as whether it is possible to compare it with other previous studies.
Author Response
Materials and method
- Review the score in the study design section.
we have corrected see the material and methods section
- It could be interesting in this section to give a summary of each of the three blocks of questions to give a general idea of ​​them.
we have corrected see the material and methods section
Results
- We recommend reviewing the names of the NGOs since some are translated into English and others are not. the same criteria should be followed we have corrected it see Lines 262-279 (table 5)
Discussion and conclusions
- As the authors have commented, the questionnaire was carried out at the beginning of the course (September 2022), so the first-year students have just entered the degree. Were the differences between newly admitted students and those who had been a year old (second-year students) assessed? The authors thank you for the question but there were no statistically significant differences between first and second year students (preclinical courses). For this reason, the authors do not mention anything about it.
- After the results obtained and the comparison with previous studies, what would be the future for this field of analysis? In our opinion, we believe that dental students (in our university) should be trained more in knowledge of global oral health, as well as the work carried out by volunteers in humanitarian dental projects.( see lines 505-511)
- Authors are recommended to review this section and include more analysis of the results obtained as well as whether it is possible to compare it with other previous studies.
we have corrected see the discussion section

Round 2
Reviewer 2 Report
Once again I would like to congratulate the authors for this study.
The authors managed to improve the text, the new discussion section is well written and documented, now supporting the results.